# Utilizing Deep Neural Networks to Fill Gaps in Small Genomes

**DOI:** 10.3390/ijms25158502

**Published:** 2024-08-04

**Authors:** Yu Chen, Gang Wang, Tianjiao Zhang

**Affiliations:** College of Computer and Control Engineering, Northeast Forestry University, Harbin 150040, China; nefu_chenyu@nefu.edu.cn (Y.C.); wanggang@nefu.edu.cn (G.W.)

**Keywords:** gap closing, gap filling, next-generation sequencing, NLP

## Abstract

With the widespread adoption of next-generation sequencing technologies, the speed and convenience of genome sequencing have significantly improved, and many biological genomes have been sequenced. However, during the assembly of small genomes, we still face a series of challenges, including repetitive fragments, inverted repeats, low sequencing coverage, and the limitations of sequencing technologies. These challenges lead to unknown gaps in small genomes, hindering complete genome assembly. Although there are many existing assembly software options, they do not fully utilize the potential of artificial intelligence technologies, resulting in limited improvement in gap filling. Here, we propose a novel method, DLGapCloser, based on deep learning, aimed at assisting traditional tools in further filling gaps in small genomes. Firstly, we created four datasets based on the original genomes of *Saccharomyces cerevisiae*, *Schizosaccharomyces pombe*, *Neurospora crassa*, and *Micromonas pusilla*. To further extract effective information from the gene sequences, we also added homologous genomes to enrich the datasets. Secondly, we proposed the DGCNet model, which effectively extracts features and learns context from sequences flanking gaps. Addressing issues with early pruning and high memory usage in the Beam Search algorithm, we developed a new prediction algorithm, Wave-Beam Search. This algorithm alternates between expansion and contraction phases, enhancing efficiency and accuracy. Experimental results showed that the Wave-Beam Search algorithm improved the gap-filling performance of assembly tools by 7.35%, 28.57%, 42.85%, and 8.33% on the original results. Finally, we established new gap-filling standards and created and implemented a novel evaluation method. Validation on the genomes of *Saccharomyces cerevisiae*, *Schizosaccharomyces pombe*, *Neurospora crassa*, and *Micromonas pusilla* showed that DLGapCloser increased the number of filled gaps by 8.05%, 15.3%, 1.4%, and 7% compared to traditional assembly tools.

## 1. Introduction

In recent decades, with the rapid development and continuous advancement of sequencing technology, the amount of sequencing data has grown exponentially, enabling the accurate sequencing of many biological genomes. Next-generation sequencing (NGS) technology is widely used due to its advantages of high speed, large data output, and high accuracy. Typical NGS data usually consist of reads that are tens to hundreds of base pairs in length. These relatively short reads are assembled into longer continuous fragments called contigs. After forming contigs, the next step is to further assemble them into larger fragments called scaffolds. However, due to issues such as low coverage areas and repetitive elements in NGS data, assembling scaffolds often results in gaps that are unknown sequences [1]. Closing these gaps in scaffolds is a crucial step in genome assembly.

Currently, many tools have been developed to fill gaps using short reads. Many genome assemblers, such as ABySS and Allpaths-LG, include gap-filling modules in their pipelines. Additionally, there are standalone tools for this purpose, such as GapCloser [2] in the SOAPdenovo package, TGS-GapCloser [3], GapFiller [4], and Gap2Seq [5]. GapCloser constructs a De Bruijn graph on the set of available reads to perform local assembly. While it is suitable for smaller genomes, its memory efficiency is very low for larger genomes [6], and it only considers read pairs with insert sizes of less than 2000 base pairs. GapFiller uses read pairs, in which one end aligns with the scaffold and the other end partially aligns with the gap regions. It then uses a k-mer-based method to assemble reads to fill the gaps. However, GapFiller only uses partial reads and does not utilize a large amount of sequence information, making it ineffective in filling long gaps.

Ref. [5] introduced a novel gap-filling computation method, in which the problem is formulated as the Exact Path Length (EPL) problem, implemented in pseudo-polynomial time through several optimizations and encapsulated in a tool named Gap2Seq. However, their method cannot scale to large genomes and fails to fill large gaps due to its computationally expensive approach.

Significant progress has been made in gap-filling tools for large genomes in recent years. Although these filling software tools can also fill small genomes, their focus is not on small genomes. Broadly speaking, large genomes (Genome Size) typically refer to genomes larger than 5 Gb, which are mostly found in gymnosperms, amphibians, and reptiles. These genomes often have very high repeat sequences and a large number of heterozygous regions. For example, the genome sizes of angiosperms (flowering plants) vary by up to 2400 times (1C = 0.063–148.8 Gb), with an average genome size of 1C = 5.7 Gb [7]. In contrast, small genomes (such as *Escherichia coli*, approximately 4.6 Mb) have higher gene density and fewer repeat sequences, making them more suitable for assembly tools using short-read sequencing technologies. However, gap filling for small genomes is equally important. Currently, there is a growing body of research on small genomes. Anjuli Meiser et al. proposed a metagenome skimming method [8], which assembled the genomes of *Evernia prunastri* and *Pseudevernia furfuracea* from whole lichen thalli metagenomic sequences using two different taxonomic binning methods. James et al. introduced an assembly algorithm, FMLRC2 [9], demonstrating its effectiveness as a de novo assembly polisher for various prokaryotic and eukaryotic genomes.

In the field of gap filling, tools such as Sealer [6] and GapPredict [10] have shown considerable utility in small genome applications. Sealer is a resource-efficient gap-filling software designed to navigate De Bruijn graphs represented by space-saving Bloom filter data structures to close gaps within scaffolds. It is reportedly scalable to gigabase-sized genomes and uses an assembly utility from the ABySS package, known as Konnector [11], as its engine to close intra-scaffold gaps. However, Sealer ignores the size differences between gaps and newly introduced sequences and does not consider the insert size of paired reads during gap filling. GapPredict is a proof-of-concept tool based on a character-level language model to predict unresolved nucleotide sequences within scaffolds. However, its network structure and prediction algorithms are overly simplistic and cannot handle the complex scenarios in gap filling. Consequently, these advanced tools still leave many gaps unfilled in the assembly of small genomes. To date, most small genome assembly tools face the following issue: They do not incorporate deep-learning technologies.

In this study, our goal is to address the challenges faced by gap-filling software when dealing with low coverage and repetitive sequences in small genomes. To achieve this, we propose DLGapCloser, a tool that leverages deep learning to generate additional assembly fragment data, aiding traditional gap-filling software in closing more gaps. Unlike conventional assembly tools, DLGapCloser utilizes deep learning to thoroughly learn gene fragments related to gap regions and predict multiple possible gap-filling outcomes. This approach enhances dataset coverage, providing traditional assembly software with richer genetic sequence information, thereby improving the gap-filling rate and accuracy of traditional tools. We tested our method on the gene sequences of *Saccharomyces cerevisiae*, *Schizosaccharomyces pombe*, *Neurospora crassa*, and *Micromonas pusilla*, comparing it with Sealer and an enhanced version of GapPredict. Experimental results indicate that our method achieves better gap-filling performance. Additionally, to explore the accuracy and completeness of the filling results, we evaluated them using QUAST [12], which confirmed that our method does not compromise the accuracy and completeness of the filling results.

The specific contributions of this study are as follows:We created new datasets based on the original genomes of *Saccharomyces cerevisiae*, *Schizosaccharomyces pombe*, *Neurospora crassa*, and *Micromonas pusilla* by supplementing them with homologous gene sequences. The original genome data sizes were 1.62 Gb, 84.04 Mb, 956.17 Mb, and 1.12 Gb, respectively. We refined these into datasets of 26.96 Mb, 72.31 Mb, 740.8 Mb, and 28.67 Mb, providing a training set for the DGCNet model.We constructed the DGCNet network model and developed a new prediction algorithm, Wave-Beam Search. The DGCNet model enhances the feature extraction capability of gene sequences and the contextual learning ability of sequences flanking gaps. The Wave-Beam Search algorithm further improves the prediction capability of the DGCNet model by avoiding premature pruning and excessive memory usage.We established a connection between deep learning and traditional assembly tools. We formulated new gap-filling standards and created and implemented a new evaluation method. We integrated deep learning with the traditional assembly tool Sealer, and experimental results show that this combination further improved Sealer’s gap-filling rate. Additionally, to adapt to the continually advancing gap-filling methods, we developed new gap-filling standards. These new standards offer more transparent and intuitive result displays and exhibit good generality across a wide range of gap-filling methods.

## 2. Results

### 2.1. Validation Results on the Dataset

DLGapCloser was evaluated on four different datasets, including *Saccharomyces cerevisiae*, *Schizosaccharomyces pombe*, *Neurospora crassa*, and *Micromonas pusilla*. We compared DLGapCloser with Sealer (accessed on 21 September 2023) and GapPredict (accessed on 22 September 2023) filling software. Sealer is designed to close gaps within scaffolds by navigating a de Bruijn graph represented by space-efficient Bloom filter data structures. It uses a succinct Bloom filter representation to fill gaps in draft assemblies. GapPredict is a proof-of-concept implementation that uses a character-level language model to predict unresolved nucleotides in scaffold gaps. We improved GapPredict to meet the evaluation standards of Sealer and DLGapCloser. We compared the filling performance of the three tools based on three metrics. Table 1 shows the performance of the three tools in terms of the number of effectively closed gaps. The gap-filling data in Table 1 is derived from the evaluation standards of Sealer and DLGapCloser. Specifically, “Gap closed” is the count filled by Sealer, while “Gap closed (100%)” and “Gap closed (90%)” are based on the evaluation standards of DLGapCloser. Detailed evaluation standards can be found in the Section 4. The last figure in Figure 1 displays the remaining number of Ns/n after filling in the draft with the three tools, and the other figures in Figure 1 visualize the data from Table 1. Additionally, we categorized the filling results and provided a brief explanation (Figure 2). Furthermore, to evaluate the quality of the filled sequences and the robustness of the methods, we compared the tools’ outputs with the reference using QUAST (accessed on 28 April 2024) [12]. We compared metrics such as mismatches, indels, and misassemblies. Overall, we observed that compared to other advanced tools, DLGapCloser was able to close a larger proportion of gap regions while keeping error lengths and misassemblies at lower levels.

For the *Saccharomyces cerevisiae* S288C dataset, as observed in Table 1, DLGapCloser filled 170, 68, and 87 gaps under the dual evaluation standards, which is higher than GapPredict’s 163, 63, and 85, and Sealer’s 148, 63, and 73. Additionally, Figure 1 shows that DLGapCloser reduced the number of Ns in the scaffold from 273 (when unfilled) to 166, which is better than the remaining 175 by GapPredict and 190 by Sealer. This indicates that our tool performs more effectively in filling gaps in small genomes.

To further validate the superiority of DLGapCloser, we conducted verification using *Schizosaccharomyces pombe*, which belongs to the same yeast family as *Saccharomyces cerevisiae* S288C. As shown in Table 1, DLGapCloser filled 109, 7, and 35 gaps, respectively, outperforming GapPredict’s 101, 7, and 32 gaps, and Sealer’s 79, 5, and 28 gaps. Figure 1 also illustrates that after filling with DLGapCloser, the number of Ns in the *Schizosaccharomyces pombe* scaffold decreased from 91 to 48, whereas GapPredict left 56 and Sealer left 68.

To assess DLGapCloser’s performance on other small genomes, we conducted additional tests on *Neurospora crassa* from the Fungi kingdom and *Micromonas pusilla* from the Viridiplantae kingdom. As seen in Table 1 and Figure 1, regardless of whether we used Sealer’s evaluation criteria or DLGapCloser’s criteria, DLGapCloser consistently outperformed both Sealer and GapPredict. This demonstrates that our tool has broad applicability and effectiveness across different small genomes.

In addition to analyzing the filling performance of various software tools from a data perspective, we also visualized the filling effects of each software using the method shown in Figure 2. We categorized the filling results into six categories: A, B, C, D, E, and F. Each category consists of four rows: The first row represents the reference genome sequence, and the remaining three rows represent the genome sequences filled by DLGapCloser, GapPredict, and Sealer, respectively. Each row is color-coded: Dark green and light green represent T and A, dark purple and light purple represent C and G, red indicates misfilled bases in the genome sequence, orange represents ambiguous or uncertain bases, black represents unfilled regions (N), and gray represents underfilled regions. In category A, both DLGapCloser and GapPredict have uncertain fillings, but DLGapCloser has fewer uncertain bases compared to GapPredict, while Sealer did not fill the gap region and even failed to identify some gap areas. In category B, DLGapCloser completely filled the gap region. Although GapPredict filled the gap, it misfilled some parts of the gap region, while Sealer left some gap regions unfilled. In categories C and D, Sealer’s filling performance was slightly better than the other two. In categories E and F, DLGapCloser completed the gap filling, whereas GapPredict and Sealer did not effectively fill the gap regions.

### 2.2. Validation of Filling Quality

Finally, Figure 3 summarizes the QUAST evaluation results across the four datasets. From the overall comparison shown in the figure, it is evident that DLGapCloser performs similarly to the other two tools in terms of mismatches and indels. For the Scaffold gap loc. mis. metric, GapPredict and DLGapCloser perform similarly to and better than Sealer. However, for the Scaffold misassemblies metric, DLGapCloser outperforms GapPredict in most datasets.

In summary, our deep learning-based method demonstrates optimal performance in terms of the number of gaps filled, misassemblies, and introduced error sequences. This means DLGapCloser can fill more gaps without generating more misassemblies and errors. DLGapCloser achieves the best or near-best performance scores across all evaluation metrics in each dataset used for evaluation, indicating that our gap-filling method is effective for small genomes.

## 3. Discussion

DLGapCloser is a gap-filling tool that uses shell scripts to extract training data from the scaffold files generated by Sealer. It trains using the DGCNet network and predicts using the Wave-Beam Search algorithm. Compared to traditional tools like Sealer and GapPredict, our tool shows improvements of 8.05%, 15.3%, 1.4%, and 7% over Sealer (calculated as the difference in the number of gaps filled by DLGapCloser and Sealer divided by the total number of gaps) and 2.2%, 4.59%, 1.32%, and 4.06% over GapPredict (calculated similarly). Additionally, we developed a new prediction algorithm, Wave-Beam Search. By using this algorithm, DLGapCloser achieved further improvements of 7.35%, 28.57%, 42.85%, and 8.33% in gap filling compared to not using the Wave-Beam Search algorithm.

Using QUAST to evaluate the performance of the three tools on four datasets (Figure 3), we found that DLGapCloser does not introduce more errors after filling gaps. This indicates that our tool can fill more gaps without causing additional errors.

However, as shown in Figure 3, the mismatch and indel indicators are generally higher. This analysis suggests that there are still repeat sequences and complex regions in the filled results, as these make correct assembly and filling difficult, easily introducing errors, increasing the complexity of filling, and leading to mismatches and insertions/deletions. In the future, we will consider using third-generation technologies, which can typically produce longer reads than second-generation technologies and are better suited for resolving complex structures and repeat sequences in genomes. Additionally, our current research is primarily focused on small genomes, such as those of eukaryotes and green algae. Our research on larger genomes, such as those of mammals, is not yet mature and may not fully represent the diversity and complexity of all genome structures.

## 4. Materials and Methods

### 4.1. DLGapCloser Algorithm

The working principle of DLGapCloser is as follows (see schematic in Figure 4), mainly divided into the following four stages. First, the data-processing stage: Using Sealer assembly software, short-read gene fragments are assembled to generate intermediate scaffold files. Gaps produced during the assembly process are identified based on the scaffold files. SAMTools (accessed on 11 April 2023) [13] and BEDtools (accessed on 11 April 2023) [14] are used to find the positional coordinates of flanking gene segments in the original genome based on gap segment information, which are then trimmed. Read data mapped to the segments in both the original genome and homologous genomes are identified, and matched read information is filtered and trimmed to form the input data required for the next step of deep learning. Second, the deep learning and sequence prediction stage: Gene sequences are converted into vector forms, and DGCNet is used to learn the spatial and sequence features of sequences. AGCT is predicted using softmax to determine the maximum possibility, and Wave-Beam search is used to obtain gap prediction data. Then, the gap-filling stage: Deep learning-predicted result data and initial short-read genome data are input into Sealer for assembly, generating new assembly results and intermediate scaffold files. Finally, the filling result evaluation stage: BWA-MEM (accessed on 12 April 2023) [15] is used to compare the gap flank sequence data generated in the data-preprocessing stage with the reference genome. The reference gene sequence of the gap is determined using coordinate information from both sides, and gap information is re-obtained from the scaffold file using the same method. Exonerate (accessed on 18 October 2023) [16] is used to compare the two sequences, and the sequence consistency percentage is used to determine the difference between the predicted assembly sequence and the reference gene sequence. At the same time, QUAST software is used to evaluate the accuracy and completeness of the filling results of the three filling tools.

### 4.2. Short-Read Datasets

To evaluate the algorithm results, we selected four biological datasets for study: *Saccharomyces cerevisiae*, *Schizosaccharomyces pombe*, *Neurospora crassa*, and *Micromonas pusilla* short-read Illumina data. In the field of gap filling, the primary focus of research has been on genomes of bacteria and humans, while there has been relatively less research on the genomes of fungi and green algae. Only a few gap-filling tools, such as GMcloser, have studied *Saccharomyces cerevisiae* S288C among fungi. To increase the diversity of our data, we chose not only *Saccharomyces cerevisiae* S288C, but also *Schizosaccharomyces pombe*, which also belongs to the phylum Ascomycota. Additionally, we selected more distantly related species, *Neurospora crassa* and *Micromonas pusilla*, to make our dataset more diverse and compelling. These data were obtained from NCBI (National Center for Biotechnology Information) and converted from SRA format to fastq format using the fastq-dump tool (accessed on 10 April 2023). Table 2 details the information about the reads used for the experiments.

### 4.3. Sealer Assembly and Gap Extraction

In this stage, the goal was to obtain the gap information generated during the assembly process. First, the initial short-read genomic sequence data was assembled using Sealer to generate an intermediate scaffold file. The scaffold file was then traversed to identify the coordinates of gaps based on the positions of N/n in the sequences. The positional information of gaps was stored in a BED file.

### 4.4. Selecting Homologous Genomes and Dataset Creation

In the DLGapCloser project, constructing training and prediction datasets is a crucial step. After determining the positions of the gaps, BEDtools was used to extend the coordinates of the gaps on both sides in the BED file to obtain genomic sequences flanking the gaps. These gene sequences served as the prediction dataset for the subsequent deep-learning model. The training dataset consisted of two parts. First, using the Bio-BloomMICategorizer (accessed on 12 April 2023) [17] tool, genomic sequences from the prediction dataset were mapped to the initial short-read genomic sequence data to generate short-read genomic data related to the gene sequences flanking the gaps, which constituted the first part of the training dataset. Secondly, based on the principle of error correction in the genome assembly process, we borrowed a method proposed by Adam M. Phillippy et al. [18], attempting to use a homologous genome that is close to the sequence information of the assembly gene as a supplementary data set and using Bio-BloomMICategorizer to map gene data flanking gaps to homologous genomes to obtain the second part of the training data.

The selection of homologous genomes mainly involved the following steps. First, we used the BLAST (accessed on 11 April 2023) tool to analyze gene fragments from the four datasets to obtain genomes with similar gene sequences. Then, through NCBI, we determined whether these similar genomes were homologous to our datasets and assessed the phylogenetic relationships. This process enabled us to obtain accurate homologous genome data.

### 4.5. Deep Learning and Gap Sequence Prediction

Gaps in the genome are primarily caused by issues such as repetitive regions and low sequencing coverage. Therefore, at this stage, we attempted to fully leverage a deep-learning framework to learn the sequence information flanking the gaps. The goal was to improve these issues through the predicted outputs.

For this paper, to better study the gene sequences flanking gaps, we divided the prediction process into three main modules: the encoding phase, network model phase, and prediction phase. In the encoding phase, we used One-hot encoding to convert DNA sequences into vector form, and the encoded vector sequences were fed into the network model for training. In the network model phase, DGCNet used CNN to extract local features from the vectorized gene sequences, then employed Bi-LSTM to learn the interrelationships between genes and the contextual information of the gene sequences. Finally, a fully connected layer was used for classification. In the prediction phase, we designed a new search algorithm, Wave-Beam Search, to predict each base of the gap one by one using the results learned by the network model.

### 4.6. Gene Sequence Encoding

In the context of genome assembly and gap filling, A represents adenine, C represents cytosine, G represents guanine, and T represents thymine. These bases are the basic units that constitute DNA molecules, forming the double-helix structure through their arrangement and pairing. During genome assembly, sequencing technologies generate sequences containing these bases, which are then assembled into complete genomes using algorithms. In the gene sequence encoding phase, this study utilized One-hot encoding to encode segments of gene sequences.

As shown in Figure 5, each base on both sides of the gap in the DNA sequence can be encoded as one of four One-hot vectors: [1, 0, 0, 0], [0, 1, 0, 0], [0, 0, 1, 0], or [0, 0, 0, 1]. Subsequently, the binary vectors of each nucleotide in the DNA sequence are merged into a binary matrix. Given a DNA sequence T = C1 C2 ⋯ Ci ⋯ CL, the One-hot encoding matrix N of sequence T can be represented as:(1)$$T=\left[u\left(M_{1}\right),u\left(M_{2}\right),\ldots ,u\left(M_{i}\right),\ldots ,u\left(M_{L}\right)\right]$$
where u is a function defined as:(2)$$u\left(M_{i}\right)=\left\{\begin{array}{c} \left(1,0,0,0\right)\;;if\;M_{i}=A\\ \left(0,1,0,0\right)\;;if\;M_{i}=C\\ \left(0,0,1,0\right)\;;if\;M_{i}=G\\ \left(0,0,0,1\right)\;;if\;M_{i}=T \end{array}\right.$$

### 4.7. DGCNet-Model

To achieve feature selection of gap gene sequence information and learn contextual information, we designed the DGCNet network architecture for this task (Figure 5). Gene sequence data is typically composed of long strings of base pairs (A, T, C, G) that exhibit high nonlinearity and complexity. Traditional feature extraction methods face many challenges when handling such high-dimensional and complex data. However, CNN’s convolutional layers are effective in capturing local patterns in sequences, making them particularly suitable for gene sequence analysis. Therefore, we used a one-dimensional convolutional neural network as the feature extraction module for the gene sequences flanking the gap. We set its output dimension to 32, used a convolution kernel size of 3, and employed ReLU as the activation function for this module. In the Bi-LSTM layer, we set the output dimension to 512 dimensions. In the fully connected layer, we classified the data into four categories, representing the probabilities of A, C, G, and T. We set a maximum of 1500 epochs, with a batch size of 64. Additionally, we implemented EarlyStopping to prevent overfitting during the training process.

In the convolutional layer, input data from the flanking segments of the gap undergo convolution with the convolution kernel, generating output feature vectors using the activation function. Equation (3) in the text describes the convolutional layer’s calculation process:(3)$$Z_{d}=t\left(W_{d}⊗X_{d}+b_{d}\right)$$
where $Z_{d}\;$ represents the output of the convolutional layer,  t denotes the Rectified Linear Unit (ReLU) activation function, $\;W_{d}$ signifies the weights, ⊗ indicates the convolution operator, $\;{\;X}_{d}\;$ denotes the input data information, and $\;b_{d}$ represents the bias value.

After convolutional operations, the extracted features may have a considerable dimensionality. Therefore, a pooling layer is introduced after the convolutional layer to effectively reduce the dimensionality of the features. This study employed a max pooling layer. Additionally, Equation (4) specifies the computational process of the max pooling operation:(4)$$Z_{q}={Pool}_{max}\left(Z_{d}\right)$$
where $Z_{q}$ represents the output of the pooling layer, and ${Pool}_{max}$ denotes the max pooling function.

Although the CNN module of the DGCNet network has achieved good results in many aspects, it primarily focuses on extracting local features and does not pay attention to the contextual information of the gene sequences, which significantly impacts the performance of subsequent gene sequence classification [19,20]. Motivated by this work, we introduced Bi-LSTM into the DGCNet network to learn the contextual features and content of DNA sequences before and after the gap. Bi-LSTM combines forward and backward LSTM units in a single model, allowing information to flow bidirectionally. This bidirectional structure enables the model to consider the sequence information before and after the current position when predicting, thus providing a more comprehensive contextual understanding. For gap filling, Bi-LSTM can accurately predict and fill missing fragments based on the sequence features on both sides of the gap, thereby improving the integrity of genome assembly. Additionally, the gating mechanism of Bi-LSTM (including input gate, forget gate, and output gate) effectively filters and memorizes critical sequence information, avoiding the vanishing and exploding gradient problems found in traditional RNNs [21]. By combining these mechanisms, Bi-LSTM can not only capture short-term dependencies, but also maintain sensitivity to long-distance dependencies. This is crucial for handling complex patterns and long-distance correlations in gene sequences. The Bi-LSTM module in the DGCNet network architecture consists of forward and backward parts to learn the features of gene sequences, and the computational formula is as follows:(5)$$g_{t}=\sigma \left(W_{a}{\cdot x}_{t}+V_{a}\cdot i_{t-1}+b_{a}\right)$$
(6)$$j_{t}=\sigma \left(W_{c}{\cdot x}_{t}+V_{b}\cdot i_{t-1}+b_{c}\right)$$
(7)$$\tilde{D_{t}}=tanh\left(W_{d}{\cdot x}_{t}+V_{d}\cdot i_{t-1}+b_{d}\right)$$
(8)$$D_{t}=g_{t}⨀D_{t-1}+j_{t}⨀\tilde{D_{t}}$$
(9)$$p_{t}=\sigma \left(W_{p}{\cdot x}_{t}+V_{p}\cdot i_{t-1}+b_{p}\right)$$
(10)$$i_{t}=p_{t}⨀tanh\;(D_{t})$$

Equation (5) represents the forget gate, which decides which gene sequence information should be discarded or retained. Equations (6) and (7) denote the input gates, used to determine which gene sequence information to update and create new candidate value vectors. Equation (8) is used to compute the current cell state. Equation (9) represents the output gate, which calculates the value of the next hidden state. Here, $W_{a}$, $W_{c}$, $W_{d}$, $W_{p}$, $V_{a}$, $V_{c}$, $V_{d},\;{and\;V}_{p}$ are weights, and $b_{a}$, $b_{c}$, $b_{d}$, and $b_{p}$ are biases.

### 4.8. Wave-Beam Search Prediction Algorithm

The Beam Search [22] algorithm is a heuristic search algorithm that explores graphs by expanding the most promising nodes in a limited set. Beam search is a modification of best-first search that reduces memory requirements. However, in the Beam Search algorithm, it adopts a greedy selection each time and retains only a predetermined number of the best partial solutions as candidates, rejecting nodes without prospects and ignoring entire branches of the decision tree grown from that node, so the best choice obtained each time is not the best choice on the entire path, which may lead to the inability to explore a wider search space and, thus, failure to achieve the global optimal solution.

Therefore, we have designed a new search algorithm called Wave-Beam Search based on Beam Search, which is different from the traditional Beam Search algorithm. It calculates the maximum probability value at the current level by continuously accumulating the probability values of each path at the current level. Due to restrictions imposed by limited memory resources, when the total number of paths is excessive, we designed an “expand–contract” pruning model, which prunes paths based on their probability values to filter out paths with lower probabilities at the current level. Therefore, we can ensure obtaining the globally optimal solution at the current level and reduce the consumption of memory resources. The Wave-Beam Search algorithm is illustrated in Figure 6.

The Wave-Beam Search algorithm mainly consists of expansion (levels 1–3 and 5–6) and contraction (levels 4 and 6) phases. In the expansion phase (levels 1–3), the algorithm retains all nodes completely due to the low memory consumption by fewer nodes in the decision tree and the lower levels of the tree. Early pruning operations would otherwise lose a large number of nodes in the lower levels. By preserving as many results as possible during this stage, compared to the Beam Search algorithm, this approach provides more possibilities and avoids the loss of high-probability nodes in subsequent stages. However, nodes are not expanded indefinitely. We can set a maximum expansion limit in the algorithm. When the number of nodes expands to or exceeds this limit, the algorithm enters the contraction phase (level 4). In the contraction phase, the Wave-Beam Search algorithm filters out partial maximum probability values based on the cumulative sum of probabilities of current-level nodes (i.e., the accumulated probabilities from the first level to the current level), discarding the rest of the nodes. Probability accumulation is a filtering strategy used in our algorithm. Since the decision tree levels gradually deepen, we only maintain the cumulative probability sum of all nodes at the current level, and we do not save the probability values of nodes before this level, thereby reducing the maintenance cost and node selection across multiple levels and nodes, as well as reducing memory usage. After the pruning in the contraction phase, the number of nodes decreases sharply, which not only releases a large amount of memory, but also retains several nodes with the highest global probability values in the decision tree, minimizing the loss of useful information. Following the contraction phase, due to the significantly reduced number of nodes, the algorithm enters the expansion phase again. This cycle of expansion and contraction phases repeats until the prediction is completed.

The Wave-Beam Search algorithm alternates between expansion and contraction phases to improve efficiency and accuracy. During the expansion phase, the algorithm retains all nodes to preserve as many possibilities as possible and avoids early pruning that would otherwise lose lower-level nodes. When the number of nodes becomes too large, the algorithm enters the contraction phase, wherein it filters out partial maximum probability values based on the cumulative sum of the node probabilities, reducing memory usage and retaining several nodes with the highest global probability values. Additionally, the algorithm uses a probability accumulation strategy to reduce maintenance costs and memory usage. The design of the Wave-Beam Search algorithm aims to improve efficiency while maintaining accuracy and minimizing the loss of useful information. We compared the traditional Beam Search algorithm with the Wave-Beam Search algorithm, and the experimental results are shown in Table 3.

### 4.9. Gap Filling

Gap filling is the most crucial step in the entire process. To enhance the filling performance of the traditional gap-filling tool Sealer and address issues such as insufficient filling due to repeated segments, reverse repeats, and low sequencing coverage, we select the top predictions from both forward and reverse predictions of each gap from the DGCNet network. We then choose complementary predictions of forward and reverse top predictions and combine these sets of predicted data with the original genomic data to reapply to Sealer for re-filling. This approach not only successfully establishes a connection between deep-learning models and traditional assembly software, but also enriches the Sealer dataset, reducing issues such as repeated segments and low sequencing coverage.

### 4.10. Evaluation of Gap Filling Results

Since the current method combines deep learning with traditional gap assembly tools, we have established new gap-filling standards and implemented a new evaluation method. In this study, we directly compare the consistency between the reference genome and the predicted genomic segments. The higher the consistency rate, the more segments the predicted sequences can match to the reference genome, indicating higher accuracy. Conversely, a lower consistency rate indicates lower accuracy. The specific evaluation steps are as follows:First, retrim the gene sequences on both sides of the gap according to the dataset creation steps. In this step, intentionally trim the gene sequences on both sides of the gap longer to improve the accuracy of the reference gene sequence in subsequent steps.Create the reference gene sequence. In this step, trim the gene sequences on both sides of the gap shorter so that the focus during sequence alignment can be on the gap prediction data rather than the gene sequence data on both sides.Use Exonerate to align the reference gene sequence and the predicted gene sequence. The files generated by the Exonerate tool after alignment need to be processed through a script, and the alignment consistency rates are sorted from 100% to 0% in descending order. In this study, we only compared cases in which the consistency rate of each assembly method was 100% and >90%.

At the same time, since gap filling is the final step in the genome assembly pipeline, errors introduced in the sequence may affect subsequent analysis, especially when the current gap region is in the coding region of the genome. Therefore, in addition to focusing on the number of gaps filled by the gap-filling tool, we also paid attention to the errors introduced by the gap-filling tool. We used QUAST to evaluate the accuracy and completeness of the results of the three gap-filling tools. The evaluation results are shown in Figure 3.

Compute server specifications: The deep learning section was conducted on the following environment:Model name: AMD Ryzen 7 5800H with Radeon Graphics 3.20 GHz (AMD, Sunnyvale, CA, USA)RAM: 16.0 GBOther sections were conducted on the following environment:Architecture: x86_64CPU(s): 384Model name: Intel(R) Xeon(R) Platinum 8260 CPU @ 2.40 GHz (Intel Corporation, Santa Clara, CA, USA)

## 5. Conclusions

Due to the high throughput and low cost of next-generation sequencing technology, researchers can obtain large-scale genomic data in a short time, providing a rich sample set for deep-learning algorithms. The advantage of deep learning lies in its ability to learn complex patterns and features from large-scale data. In genomics, deep learning is widely used for tasks such as sequence analysis, variant detection, and functional annotation. However, in the field of gap filling, deep learning has not been well utilized due to the scarcity of training data and the substantial computational resources required.

Based on the above background, we proposed DLGapCloser, which utilizes deep learning to learn features and patterns in sequencing data to predict gap gene sequences, improving the accuracy and coverage of traditional filling software data and further enhancing the assembly accuracy and quantity of traditional software. By comparing with Sealer and GapPredict, we found that DLGapCloser has the following advantages: First, we used homologous genomes to expand the training data, providing more possibilities for subsequent learning and prediction. Additionally, we employed deep learning to learn the features and contextual content of gap gene sequences and constructed a complex network structure with detailed parameter tuning. Third, we designed the Wave-Beam Search algorithm, which broadens the prediction scope and improves the accuracy of the predicted data. Finally, DLGapCloser offers new possibilities for overcoming the limitations of traditional methods in the field of gap filling. Although our software has achieved good results in small genomes, it has not yet achieved ideal results in large genomes. We will continue to optimize the tool to improve the effectiveness and accuracy of gap filling in large genomes.

## Figures and Tables

**Figure 1 ijms-25-08502-f001:**
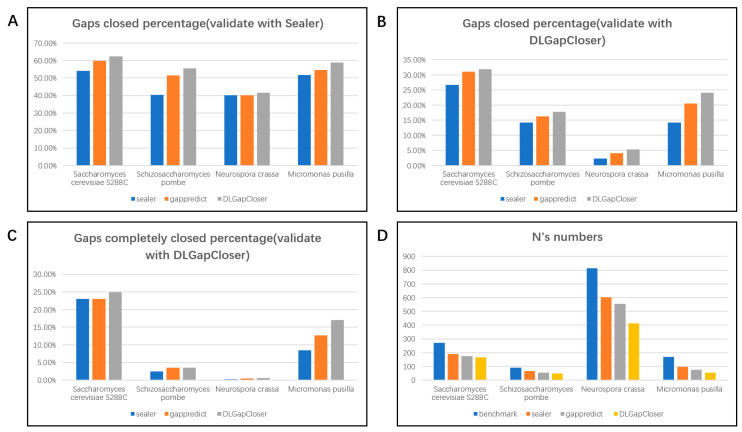
Gap-filling improvements and effects on the draft assemblies. (**A**) shows the percentage of filled gaps for three tools across four datasets, evaluated using the Sealer evaluation standard; (**B**) shows the percentage of filled gaps for three tools across four datasets, evaluated using the DLGapCloser evaluation standard; (**C**) shows the percentage of completely filled gaps for three tools across four datasets, evaluated using the DLGapCloser evaluation standard; (**D**) shows the remaining number of N/n on the filled scaffolds after gap filling by the three tools.

**Figure 2 ijms-25-08502-f002:**
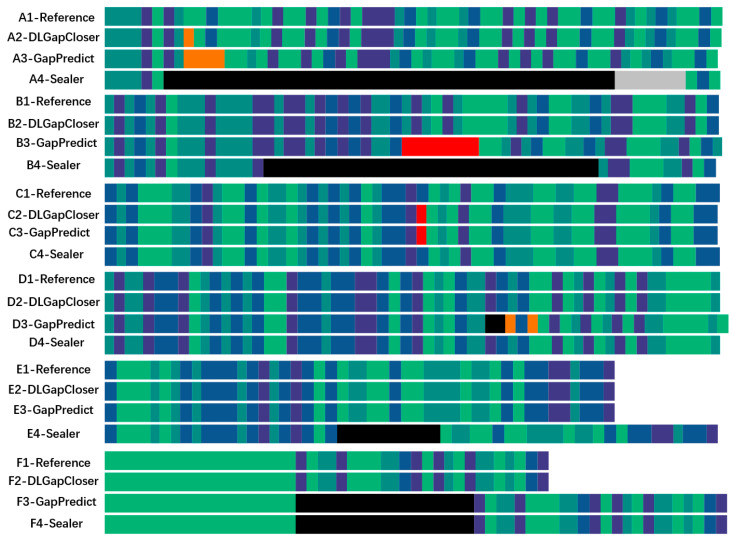
Visualization of various gap-filling results. A–F represent six types of gap-filling categories selected from the filling results. Each filling category is composed of four rows: The first row represents the reference gene sequence, and the remaining three rows represent the gene sequences filled by DLGapCloser, GapPredict, and Sealer, respectively.

**Figure 3 ijms-25-08502-f003:**
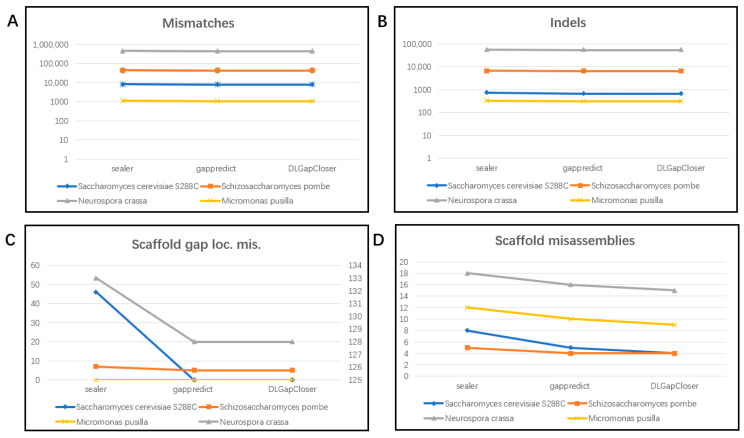
Quality of the gap-filled assemblies of various genomes using various tools. (**A**) shows the number of mismatches in the filling results of the three tools across four datasets using the QUAST tool; (**B**) shows the number of indels in the filling results of the three tools across four datasets using the QUAST tool; (**C**) shows the number of positions in the scaffolds (breakpoints) wherein the flanking sequences are combined at the wrong distance in the scaffold (causing a local misassembly) in the filling results of the three tools across four datasets using the QUAST tool; (**D**) shows the number of misassemblies at the scaffold level in the filling results of the three tools across four datasets using the QUAST tool.

**Figure 4 ijms-25-08502-f004:**
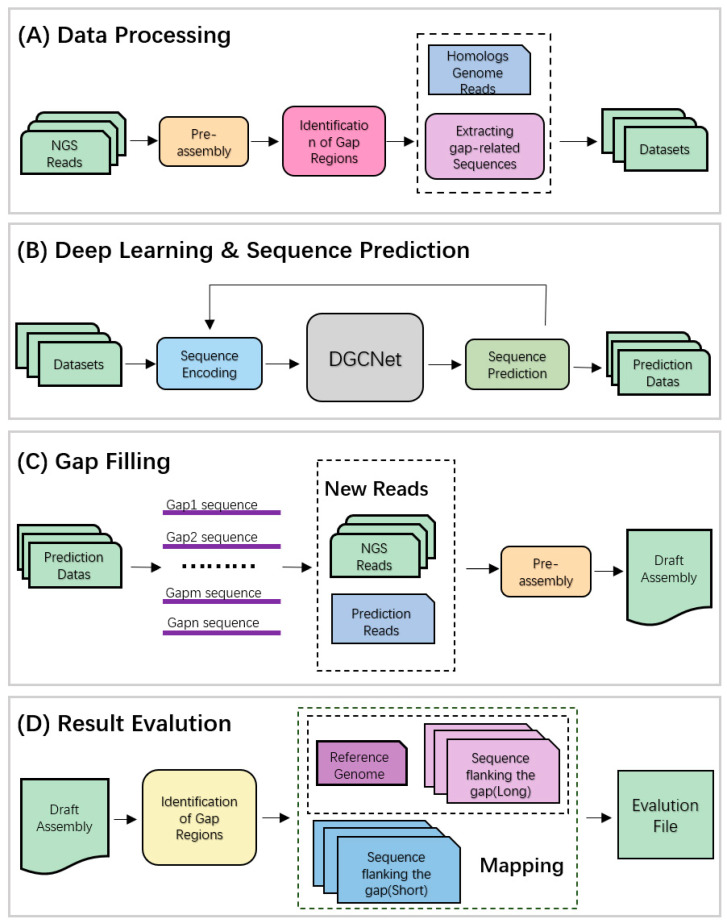
Schematic overview of the DLGapCloser algorithm. (**A**) Data processing; (**B**) DGCNet network learning and prediction combined with Wave-Beam Search algorithm; (**C**) Gap filling; (**D**) Result evaluation.

**Figure 5 ijms-25-08502-f005:**
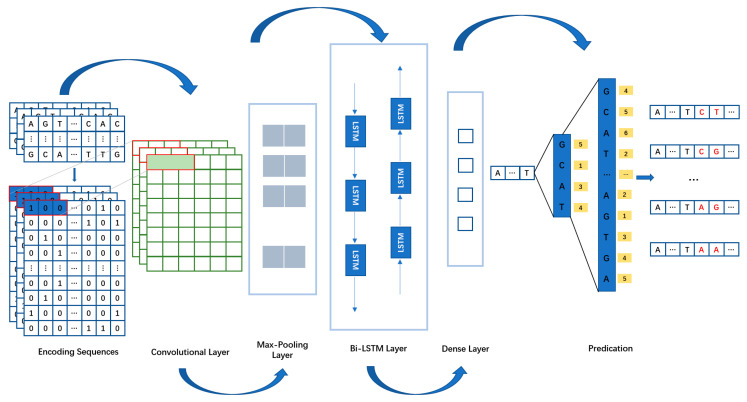
A generic architecture of DGCNet. Consists of the encoding sequences part, the network architecture part (Convolutional Layer, Max-Pooling Layer, Bi-LSTM Layer, and Dense Layer), and the prediction part (Wave-Beam Search). The red letters represent newly generated gene sequences.

**Figure 6 ijms-25-08502-f006:**
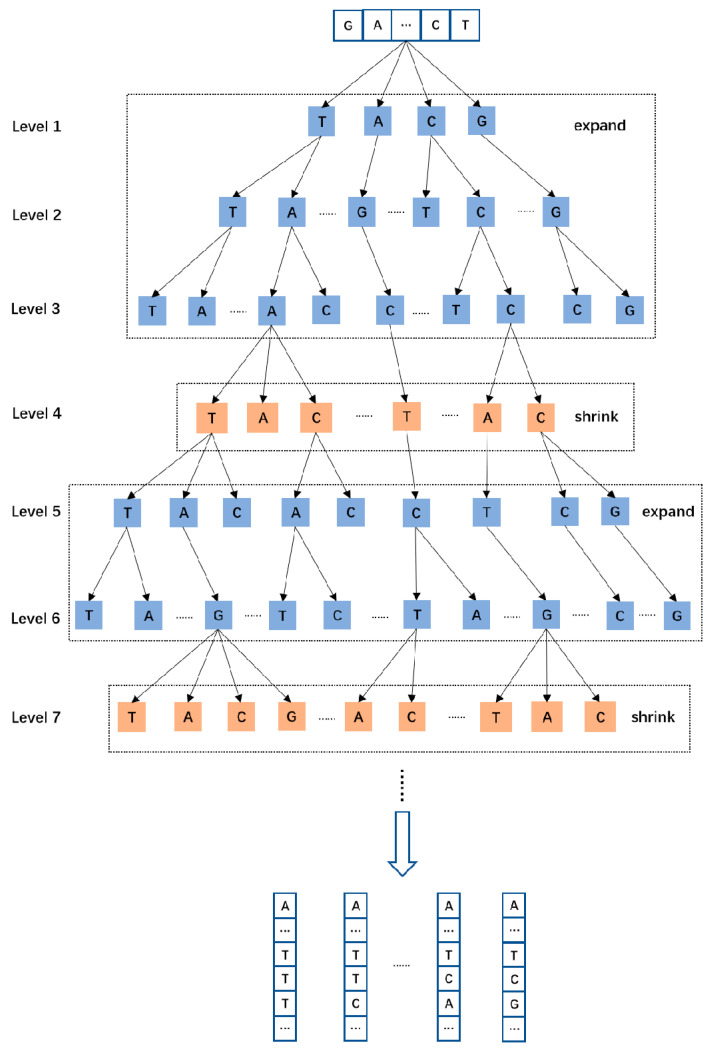
Overview of the Wave-Beam search algorithm. The algorithm is divided into the Expand phase (Levels 1–3 and Levels 5–6) and the Shrink phase (Level 4 and Level 7). The two phases alternate, and the algorithm ends the prediction when the predicted length is reached.

**Table 1 ijms-25-08502-t001:** Gap closure results.

Method	Sealer	GapPredict	DLGapCloser
*Saccharomyces cerevisiae* S288C			
Gap count	273	273	273
Gap closed	148	163	170
*Schizosaccharomyces pombe*			
Gap count	196	196	196
Gap closed	79	101	109
*Neurospora crassa*			
Gap count	1207	1207	1207
Gap closed	484	485	501
*Micromonas pusilla*			
Gap count	141	141	141
Gap closed	73	77	83

**Table 2 ijms-25-08502-t002:** Dataset information.

Dataset	Homologs Number	Accession Number	Refseq	#Bases
*Saccharomyces cerevisiae*	SRR23920092	ERR156523	GCA_000146045.2	316.5M
*Schizosaccharomyces pombe*	SRR26143067	ERR9706986	GCF_000002945.1	1.6G
*Neurospora crassa*	ERR11413973	SRR19285165	GCF_000182925.2	2.9G
*Micromonas pusilla*	SRR14462310	SRR14462380	GCF_000090985.2	1.6G

**Table 3 ijms-25-08502-t003:** Gap closure results using two algorithm.

Algorithm	Beam Search	Wave-Beam Search
*Saccharomyces cerevisiae* S288C	68	73
*Schizosaccharomyces pombe*	7	9
*Neurospora crassa*	7	10
*Micromonas pusilla*	24	26

## Data Availability

The data and software that support the findings of this study are available from the corresponding author upon reasonable request.

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
