# Peer review of "Utilizing Deep Neural Networks to Fill Gaps in Small Genomes"

_ijms, 2024, doi:10.3390/ijms25158502_

Round 1

Reviewer 1 Report

Comments and Suggestions for Authors

Review of the article "Utilizing Deep Neural Networks to Fill Gaps in Compact Genomes"

In my opinion, the article is interesting and potentially valuable.

My points for correcting the article:

1. The work is dedicated to compact genomes. The term "compact genome" appears only in the title of the article, and for this reason, it is unclear what the Authors mean by using this term. The limitation of the method presented in the article is that it can only be used for compact genomes.

In the Introduction, more information should be presented regarding compact genomes, along with a discussion on the limitations of the presented method in the Discussion section. Moreover, the reasons for these limitations should be clearly stated.

2. In the Author Contributions statement (page 14), there is: "... software, G.W.; ...", but in the article, only very laconic information is presented about this software, i.e., more detailed information has to be added about the software implementation by the Authors, including the programming language used for this implementation.

3. Implementation of DLGapCloser can be considered the aim and an achievement of the work (lines 83-86: "In this study, our goal is to address the challenges faced by gap-filling software when dealing with low coverage and repetitive sequences in small genomes. To achieve this, we propose DLGapCloser, a tool that leverages deep learning to generate additional assembly fragment data, aiding traditional gap-filling software in closing more gaps"), but I am afraid that the information presented in the article is insufficient for the readers of this article to implement the proposed method.

Therefore, it would be advisable for the Authors to provide the possibility for readers to download the implemented method (i.e., DLGapCloser), for example, as a *.dll or *.class file.

4. In the Introduction is (lines 62,63):

"Significant progress has been made in gap-filling tools for large genomes in recent years. However, gap-filling for small genomes is equally important."

In the Introduction, there should be added information on what a large genome and a small genome mean (what is the threshold), along with information on why software used for large genomes cannot be used for small genomes.

5. On page 10, the encoding system (2) and rules are presented:

A = "1,0,0,0"

C = "0,1,0,0"

G = "0,0,1,0"

T = "0,0,0,1"

The Encoding Sequences that are presented in Figure 5 do not fit this encoding system, for example, in Figure 5, C is encoded as "0,0,1,0".

6. More information has to be added regarding the deep learning technique implemented by the Authors, along with a justification for why they decided to use deep learning in this work.

7. An artificial neural network has been used by the Authors, but almost nothing is known about this neural network.

Because the term "neural networks" appears, among others, in the title of the article ("Utilizing Deep Neural Networks to Fill Gaps in Compact Genomes"), for this reason, more information has to be added about the neural network implemented in the work. Information about the neural network should include, among others, the number of inputs, the number of outputs, the number of hidden layers, the number of neurons in the hidden layer(s), the artificial neural network training method, training error, momentum, learning rate, time of training.

8. In the Abstract is:

"Experimental results showed that the Wave-Beam Search algorithm improved the gap-filling performance of assembly tools by 7.35%, 28.57%, 42.85%, and 8.33% on the original results. Finally, we established new gap-filling standards and created and implemented a novel evaluation method. Validation on the genomes of Saccharomyces cerevisiae, Schizosaccharomyces pombe, Neurospora crassa, and Micromonas pusilla showed that DLGapCloser increased the number of filled gaps by 8.05%, 15.3%, 1.4%, and 7% compared to traditional assembly tools."

The values presented in the Abstract: "7.35%, 28.57%, 42.85%, and 8.33%" and "8.05%, 15.3%, 1.4%, and 7%" occur only in the Abstract, and it is not known what their genesis is.

9. The Discussion section presented in the article (pages 6, 7) can be considered Conclusions rather than Discussion. For this reason, there is almost a lack of discussion in the article. Especially to emphasize the correctness and importance of the results, the discussion part should contain a comparison of the effectiveness of the proposed method with the effectiveness of the existing methods (i.e., there can be added a more extended discussion of the results in comparison with, for example, the traditional Beam Search algorithm).

The values presented in the Abstract (i.e., "7.35%, 28.57%, 42.85%, and 8.33%" and "8.05%, 15.3%, 1.4%, and 7%") should also be explained and extensively discussed in the Discussion section.

10. The figure descriptions are very short. The figure descriptions should be extended to more clearly explain the content of the figures.

Author Response

Comments 1: The work is dedicated to compact genomes. The term "compact genome" appears only in the title of the article, and for this reason, it is unclear what the Authors mean by using this term. The limitation of the method presented in the article is that it can only be used for compact genomes.

In the Introduction, more information should be presented regarding compact genomes, along with a discussion on the limitations of the presented method in the Discussion section. Moreover, the reasons for these limitations should be clearly stated.

Response 1: Thank you for pointing this out. We agree with this comment. Therefore, We have changed "Compact Genomes" to "Small Genomes" in the title on the first page. Thank you again for your thorough review and valuable comments.

Comments 2:In the Author Contributions statement (page 14), there is: "... software,G.W.; ...", but in the article, only very laconic information is presented about this software, i.e., more detailed information has to be added about the software implementation by the Authors, including the programming language used for this implementation.

Response 2: Thank you for pointing this out. We agree with this comment. We have made the Python and shell code for DLGapCloser publicly available at https://github.com/AndroidBirdBoom/DLGapCloser.git. Additionally, We have updated the code address in line 31 of the abstract section on the first page of the manuscript.

Comments 3: Implementation of DLGapCloser can be considered the aim and an achievement of the work (lines 83-86: "In this study, our goal is to address the challenges faced by gap-filling software when dealing with low coverage and repetitive sequences in small genomes. To achieve this, we propose DLGapCloser, a tool that leverages deep learning to generate additional assembly fragment data, aiding traditional gap-filling software in closing more gaps"), but I am afraid that the information presented in the article is insufficient for the readers of this article to implement the proposed method.

Therefore, it would be advisable for the Authors to provide the possibility for readers to download the implemented method (i.e., DLGapCloser), for example, as a *.dll or *.class file.

Response 3: Thank you for pointing this out. We agree with this comment. We have made the Python and shell code for DLGapCloser publicly available at https://github.com/AndroidBirdBoom/DLGapCloser.git. Additionally, We have updated the code address in line 31 of the abstract section on the first page of the manuscript.

Comments 4: In the Introduction is (lines 62,63):

"Significant progress has been made in gap-filling tools for large genomes in recent years. However, gap-filling for small genomes is equally important."

In the Introduction, there should be added information on what a large genome and a small genome mean (what is the threshold), along with information on why software used for large genomes cannot be used for small genomes.

Response 4: Thank you for pointing this out. We agree with this comment. We referred to the article by Steven Dodsworth et al., "Genome size diversity in angiosperms and its influence on gene space" (https://doi.org/10.1016/j.gde.2015.10.006), to differentiate between large and small genomes based on whether the Genome Size exceeds 5 Gb. More details have been added to lines 62-71 of the introduction section on the second page.

Comments 5: On page 10, the encoding system (2) and rules are presented:

A = "1,0,0,0"

C = "0,1,0,0"

G = "0,0,1,0"

T = "0,0,0,1"

The Encoding Sequences that are presented in Figure 5 do not fit this encoding system, for example, in Figure 5, C is encoded as "0,0,1,0".

Response 5: Thank you for pointing this out. We agree with this comment. We have readjusted the coding regions in Figure 5 on page 9, line 316, to correspond with the coding system.

Comments 6: More information has to be added regarding the deep learning technique implemented by the Authors, along with a justification for why they decided to use deep learning in this work.

Response 6: Thank you for pointing this out. We agree with this comment. We have added some deep learning parameter information in Section 4.7. DGCNet-Model on pages 10, lines 356-360. Additionally, we have included the advantages of using deep learning with DLGapCloser in the newly added Section 5. Conclusions on pages 15, lines 525-536. We have also updated the code for the implemented deep learning techniques in line 31 of the abstract section of the manuscript to facilitate reader access.

Comments 7: An artificial neural network has been used by the Authors, but almost nothing is known about this neural network.

Because the term "neural networks" appears, among others, in the title of the article ("Utilizing Deep Neural Networks to Fill Gaps in Compact Genomes"), for this reason, more information has to be added about the neural network implemented in the work. Information about the neural network should include, among others, the number of inputs, the number of outputs, the number of hidden layers, the number of neurons in the hidden layer(s), the artificial neural network training method, training error, momentum, learning rate, time of training.

Response 7: Thank you for pointing this out. We agree with this comment. We have added some new information about the model training in Section 4.7. DGCNet-Model on pages 10, lines 356-360. Additionally, we have hosted more detailed model training information in the code repository on GitHub and updated the code address in line 31 of the abstract section on the first page of the manuscript.

Comments 8: In the Abstract is:

"Experimental results showed that the Wave-Beam Search algorithm improved the gap-filling performance of assembly tools by 7.35%, 28.57%, 42.85%, and 8.33% on the original results. Finally, we established new gap-filling standards and created and implemented a novel evaluation method. Validation on the genomes of Saccharomyces cerevisiae, Schizosaccharomyces pombe, Neurospora crassa, and Micromonas pusilla showed that DLGapCloser increased the number of filled gaps by 8.05%, 15.3%, 1.4%, and 7% compared to traditional assembly tools."

The values presented in the Abstract: "7.35%, 28.57%, 42.85%, and 8.33%" and "8.05%, 15.3%, 1.4%, and 7%" occur only in the Abstract, and it is not known what their genesis is.

Response 8: Thank you for pointing this out. We agree with this comment. In the revised manuscript, we have provided the detailed calculation methods for “7.35%, 28.57%, 42.85%, and 8.33%” and “8.05%, 15.3%, 1.4%, and 7%” on pages 6, lines 222-228, in Section 3. Discussion.

Comments 9: The Discussion section presented in the article (pages 6, 7) can be considered Conclusions rather than Discussion. For this reason, there is almost a lack of discussion in the article. Especially to emphasize the correctness and importance of the results, the discussion part should contain a comparison of the effectiveness of the proposed method with the effectiveness of the existing methods (i.e., there can be added a more extended discussion of the results in comparison with, for example, the traditional Beam Search algorithm).

The values presented in the Abstract (i.e., "7.35%, 28.57%, 42.85%, and 8.33%" and "8.05%, 15.3%, 1.4%, and 7%") should also be explained and extensively discussed in the Discussion section.

Response 9: Thank you for pointing this out. We agree with this comment. We have revised the original content of Section 3. Discussion and relocated it to Section 5. Conclusions, which can now be found on pages 15, lines 518-539 of the manuscript. Additionally, we have rewritten Section 3. Discussion, which is now presented on pages 6, lines 220-241. The updated content has been incorporated into the newly submitted manuscript.

Comments 10: The figure descriptions are very short. The figure descriptions should be extended to more clearly explain the content of the figures.

Response 10: Thank you for pointing this out. We agree with this comment. We have added more descriptive information below each figure and table, and the content has been updated in the newly submitted manuscript.

Reviewer 2 Report

Comments and Suggestions for Authors

The paper "Utilizing Deep Neural Networks to Fill Gaps in Compact Genomes" presents a novel approach, DLGapCloser, for improving gap-filling in small genome assemblies using deep learning techniques. The method's integration of homologous genomes and the introduction of the Wave-Beam Search algorithm are noteworthy innovations. The study demonstrates promising results across four different organisms. However, the paper could benefit from a more detailed explanation of figure contents, a broader discussion on the method's applicability to diverse genome types, and a clearer justification for the chosen datasets. Additionally, addressing the potential biases in using homologous genomes and exploring the interpretability of the deep learning model would strengthen the study. Elaborating on these aspects would enhance the paper's impact and provide a more comprehensive understanding of it’s capabilities and limitations in the field of genome assembly.

Line 132: Clearly explain what each plot in Figure 1 represents, rather than vaguely stating "other figures visualize Table 1 data". This will help readers better understand and interpret the charts.

Line 161. Concluding "broad applicability" based on just a few small genomes may be hasty. Discuss potential limitations of DLGapCloser on different genome types or sizes, or propose plans for testing on more diverse genomes.

Line 202: Briefly explain why deep learning is less applied in this field, and highlight DLGapCloser's innovative significance.

Line 254: Explain the rationale behind selecting these four biological datasets. Are they representative? Do they cover different genome structures and complexities? Clarify how these datasets reflect various genome gap-filling challenges.

Line 275. Using homologous genomes as supplementary data is interesting but may introduce bias. Discuss how to select appropriate homologous genomes and handle differences between homologous and target genomes. Consider using multiple homologous genomes or those with varying evolutionary distances to assess method robustness.

Line 291: Please consider explaining the choice of this specific network structure combination. Discuss how different structures (pure CNN, RNN, or transformer) might affect performance. Consider using attention mechanisms to improve long-range dependency capture.

Other comments: Deep learning models are often criticized as "black boxes". Consider increasing model interpretability. For example, analyze features extracted by CNN or observe LSTM unit activation patterns to understand how the model learns and predicts gene sequences. This could enhance model credibility and potentially provide new genomic insights.

Author Response

Comments 1: Line 132: Clearly explain what each plot in Figure 1 represents, rather than vaguely stating "other figures visualize Table 1 data". This will help readers better understand and interpret the charts.

In the Introduction, more information should be presented regarding compact genomes, along with a discussion on the limitations of the presented method in the Discussion section. Moreover, the reasons for these limitations should be clearly stated.

Response 1: Thank you for pointing this out. We agree with this comment. We have added more detailed descriptions below Figure 1 and other figures. You can find the updated content in the newly submitted manuscript.

Comments 2: Line 161. Concluding "broad applicability" based on just a few small genomes may be hasty. Discuss potential limitations of DLGapCloser on different genome types or sizes, or propose plans for testing on more diverse genomes.

Response 2: Thank you for pointing this out. We agree with this comment. We removed the term "broad applicability" and specified the scope of DLGapCloser in section 3, Discussion, on pages 6, lines 238-241, to clarify its applicability and current limitations for readers.

Comments 3: Line 202: Briefly explain why deep learning is less applied in this field, and highlight DLGapCloser's innovative significance.

Response 3: Thank you for pointing this out. We agree with this comment. In the newly added section 5. Conclusions on pages 15, lines 523-524, we briefly explain why deep learning has not been widely used in the gap-filling domain. Additionally, lines 529-539 highlight the advantages of DLGapCloser and outline our future research directions. We are pleased to offer new possibilities for addressing the limitations of traditional methods in the gap-filling field.

Comments 4: Line 254: Explain the rationale behind selecting these four biological datasets. Are they representative? Do they cover different genome structures and complexities? Clarify how these datasets reflect various genome gap-filling challenges.

Response 4: Thank you for pointing this out. We agree with this comment. We have updated the reasons for selecting these four datasets in section 4.2. Short-read datasets on pages 8, lines 274-281. These datasets include fungi and algae, which possess different genomic structures and certain levels of complexity. Currently, most tools have not been studied in these categories. Therefore, we aim to address the research gap in this field by performing gap-filling and analysis on the aforementioned four datasets.

Comments 5: Line 275. Using homologous genomes as supplementary data is interesting but may introduce bias. Discuss how to select appropriate homologous genomes and handle differences between homologous and target genomes. Consider using multiple homologous genomes or those with varying evolutionary distances to assess method robustness.

Response 5: Thank you for pointing this out. We agree with this comment. We have added information on how to select homologous genomes in section 4.4. Dataset creation on pages 9, lines 306-310. Regarding the suggestion to include homologous genomes with varying evolutionary distances, we have conducted some research on different homologous genomes. However, due to resource and time constraints, we selected specific homologous genomes to ensure internal consistency and comparability within the study.

Comments 6: Line 291: Please consider explaining the choice of this specific network structure combination. Discuss how different structures (pure CNN, RNN, or transformer) might affect performance. Consider using attention mechanisms to improve long-range dependency capture.

Response 6: Thank you for pointing this out. We agree with this comment. We have re-explained the specific responsibilities of each module in section 4.5. Deep Learning and Gap Sequence Prediction, on pages 10, lines 322-329. We experimented with using only CNN or LSTM models; although these combinations performed better in terms of time, the results were unsatisfactory. Consequently, we chose the optimal CNN+Bi-LSTM combination for comparison with Sealer and GapPredict. In our previous research, we also tried a network architecture combining CNN+Bi-LSTM+Self-Attention Mechanism. However, the computational and memory overhead led us to abandon this approach, as the improvement in gap filling was minimal. Considering time and resource constraints, we decided to remove this module.

Comments 7: Other comments: Deep learning models are often criticized as "black boxes". Consider increasing model interpretability. For example, analyze features extracted by CNN or observe LSTM unit activation patterns to understand how the model learns and predicts gene sequences. This could enhance model credibility and potentially provide new genomic insights.

Response 7: Thank you for pointing this out. We agree with this comment. We updated the specific responsibilities of each module in section 4.5, Deep Learning and Gap Sequence Prediction, on pages 10, lines 322-329, and added more information about the network architecture in section 4.7, DGCNet-Model, on pages 10, lines 356-360. Additionally, we synchronized the code address to the abstract section on page 1, line 31, to facilitate reader access.

Round 2

Reviewer 1 Report

Comments and Suggestions for Authors

The Authors addressed all of my points correctly and thoroughly. Now the article is better, and in my opinion, it can be published in IJMS.